# PIPer: On-Device Environment Setup via Online Reinforcement Learning

## Abstract

Environment setup—the process of configuring the system to work with a specific software project—represents a persistent challenge in Software Engineering (SE). Automated environment setup methods could assist developers by providing fully configured environments for arbitrary repositories without manual effort. This also helps SE researchers to scale execution-based benchmarks. However, recent studies reveal that even state-of-the-art Large Language Models (LLMs) achieve limited success in automating this task. To address this limitation, we tune a specialized model for environment setup. We combine supervised fine-tuning for generating correct Bash scripts and Reinforcement Learning with Verifiable Rewards (RLVR) to adapt it to the task of environment setup. On EnvBench-Python, our method enables Qwen3-8B (a model runnable on consumer hardware) to perform on par with larger models—Qwen3-32B and GPT-4o. The training code and model checkpoints are available online: https://github.com/PIPer-iclr/PIPer.

## 1 Introduction

Large Language Models (LLMs) show great promise for Software Engineering (SE) tasks (Liu et al., 2024). While closed-source general-purpose models largely dominate benchmarks (Jain et al.; Jimenez et al., 2024), open-source models remain strong competitors (DeepSeek-AI, 2025; Qwen Team, 2025; Kimi Team et al., 2025). Recent studies demonstrate that task-specific autonomous agents powered by open-source models can solve various SE problems, including code generation (Hasan et al., 2025), bug localization (Ma et al., 2025; Chang et al., 2025; Reddy et al., 2025; Chen et al., 2025b), and issue resolution (Luo et al., 2025; Wang, 2025; Pan et al., 2025; Zeng et al., 2025; Ma et al., 2025; Chang et al., 2025).

A common strategy for developing capable task-specific agents is to train them on carefully curated datasets (Pan et al., 2025; Zeng et al., 2025). However, in the SE domain, the bottleneck has shifted from sophisticated data filtering strategies to acquiring sufficient data in the first place. Since agents operate in an interactive manner, this requires scaling the construction of interactive environments. This, in turn, often requires appropriately configuring the system to be able to execute the sample code. In this paper, we will call this configuration process an environment setup.

This limitation has far-reaching implications for SE benchmarks. For instance, SWE-Bench (Jimenez et al., 2024), one of the leading benchmarks for SE agents, includes only 12 Python repositories, and collecting and maintaining it required substantial manual effort. Scaling such datasets typically relies on manual setup (Pan et al., 2025) or on synthetic augmentation (Pham et al., 2025), trading realism for scale. Automated environment setup methods (Guo et al., 2025; Badertdinov et al., 2025; Zhang et al., 2025; Vergopoulos et al., 2025) promise scalability with real data but remain limited—for instance, SWE-Rebench (Badertdinov et al., 2025) reports a $31\%$ success rate on Python repositories overall, while on EnvBench (Eliseeva et al., 2025), a recently introduced benchmark for environment setup specializing on hard repositories, the best result is $6.69\%$ (22 out of 329), achieved by GPT-4o in an agentic workflow.

We seek to improve small open-source models to democratize the usage of LLMs for environment setup. To this end, we analyze the environment setup scripts produced by strong LLMs on EnvBench and employ both supervised fine-tuning (SFT) and reinforcement learning (RL) to resolve found issues. The proposed method achieves more than $9\times$ improvement over the base model, being on par with the open-source model four times the size, and strong closed-source baselines. Specifically, our

contributions are: (1) the first application of online reinforcement learning with lightweight verifiable reward to environment setup, (2) on-device sized PIPER model performing on par with strong baselines offering a superior performance-cost ratio, and (3) a rigorous evaluation, demonstrating that the model trained with the proposed method generalizes across different datasets, indicating genuine scripting capability enhancement. To facilitate reproducibility and future research in this direction, we make our code, model weights, and generated scripts publicly available [1].

The rest of the manuscript is organized as follows. We describe the datasets used for training and evaluation in Section 2, motivate and describe the training approach in Section 3, describe how we set up the experiments in Section 4, and provide an overview of our experimental results in Section 5.

## 2 DATASET

The focus of our work is to democratize the use of LLMs for environment setup. To measure our progress in this task, we select two environment setup benchmarks, EnvBench (Eliseeva et al., 2025) and Repo2Run (Hu et al., 2025b). Also, to check how this training affects a broader set of tasks, we employ Terminal-Bench (The Terminal-Bench Team, 2025). In this section, we outline the specifics of each dataset we use—the inputs and outputs of the evaluated method, and the definition of task resolution.

**EnvBench-Python** comprises 329 Python repositories from GitHub. As an input, an environment setup approach has access to the full repository context and base environment configuration. How exactly this context is utilized remains part of the approach definition: it could be a predefined prompt, an interactive agentic workflow, and more. As an output, an environment setup approach should produce a shell script that installs all the needed dependencies in the base environment. The correctness of the environment setup script is evaluated by first executing it, and then invoking Pyright[2]—a static analysis tool used to evaluate whether the imports across the codebase were resolved successfully. The repository is considered to be set up correctly if the script finished with exit code 0 and subsequent Pyright check reported no import issues.

**Repo2Run** comprises 420 Python repositories from GitHub with no overlaps with EnvBench-Python. The original work primarily focuses on an agentic setting, where an environment setup agent is granted access to the base environment with the repository through a terminal interface and other specialized tools. The agent is then expected to autonomously configure the repository by interacting with the environment. In contrast with static analysis-based metrics from EnvBench-Python, Repo2Run runs test collection via pytest[3] to verify the environment setup correctness. We include Repo2Run to verify that our experimental results transfer across different repositories and success criteria. We additionally adapt Repo2Run to settings beyond agentic, employing a more general task formulation similar to that of EnvBench-Python (discussed in detail in Section 4.2).

**Terminal-Bench** comprises 80 tasks focused on command-line environment configuration tasks (we use version 0.1.1 of the benchmark), evaluating AI agents' ability to handle real-world, end-to-end terminal operations, including compiling code, training models, and setting up servers. Each task consists of the problem described in natural language passed to an LLM, a Docker environment, and a test script to verify if the agent completed the task successfully. We use the original implementation[4] with multi-turn agentic scaffold Terminus 1. The success is determined by whether the agent can complete the specified terminal-based objective within a sandbox environment. We use Terminal-Bench to assess whether our training pipeline, designed primarily for single-turn Python package installation scenarios, generalizes to broader, out-of-distribution, multi-turn terminal command execution tasks beyond dependency management.

## 3 METHOD

To train the model, we employ a two-stage process widely adopted in the literature (Liu et al., 2025b; Yoshihara et al., 2025; Golubev et al., 2025). First, we tune the model in a supervised manner on the

---

[1]Replication Package: `https://github.com/PIPer-iclr/PIPer`

[2]`https://microsoft.github.io/pyright`

[3]`https://pytest.org`

[4]`https://github.com/laude-institute/terminal-bench`

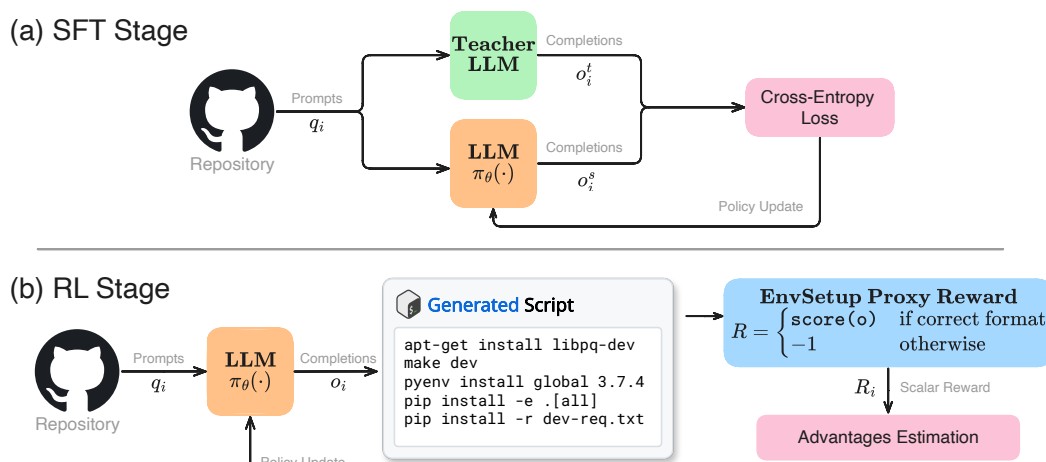

Figure 1: Overview of the proposed training pipeline. (a) **SFT training**: For the $i$-th sample (a repository), both teacher and student LLMs receive the prompt $q_i$, which includes the task description and repository context. They generate completions $o_i^t$ and $o_i^s$, respectively, expected to contain a shell script. The student model's weights are updated by minimizing the cross-entropy loss between its output distribution and the teacher's completion. (b) **RL training**: For each sample, LLM $\pi_\theta$ generates a completion $o_i$, expected to contain a shell script. The completion is evaluated by a rule-based reward function $R$, which outputs a score $R_i$. The REINFORCE++ algorithm then updates the LLM weights using the rewards $R_i$ and responses $o_i$.

executable scripts sampled from the larger model of the same family. Then, we run one more stage of RL training, to refine the capabilities of the model after the SFT update. We employ the RLVR technique since it has been reported to show promising results on tasks from the SE domain (Luo et al., 2025; Golubev et al., 2025). In our notation throughout this paper, we use $q$ to denote prompts provided to the model, $o$ to represent model responses, $s$ to refer to shell scripts extracted from model outputs, and $\pi_\theta$ to denote the model with parameters $\theta$. We use regular expressions to extract the shell script from the model outputs, and if parsing fails, we consider $s$ to be empty. The schematic representation of the training stages is illustrated in Figure 1. Further, we introduce the details of the method. In Section 3.1 we discuss the SFT training, and in Section 3.2 we introduce the RL training.

### 3.1 SUPERVISED FINE-TUNING

The supervised fine-tuning involves training a model on a set of data points that are considered to be ground truth. However, for the task of environment setup, and for the hard repositories specifically, it is a costly task to obtain such ground truth scripts. The authors of EnvBench provide only a small number of scripts generated by experts, and even strong models are solving only a small portion of the dataset (Eliseeva et al., 2025). Due to this, we employ distillation (Hinton et al., 2015), a technique where the small model (called Student) learns to imitate the behavior of a larger model (called Teacher). Our setup is shown in Figure 1(a) and detailed below.

We implement the SFT stage using executable scripts collected during the evaluation of a larger Qwen3-32B model. We first collect samples $\{q_i, o_i^t\}$ from evaluation rollouts. Then we filter out the samples where $o_i^t$ doesn't contain a script, or the script results in a non-zero exit code. Finally, we select 2,500 pairs $\{q_i, o_i^t\}$ at random to form the distillation dataset. The student model $\pi_\theta$ is trained on this dataset in a supervised manner without further changes or masking. Since these samples originate from a different, larger model rather than $\pi_\theta$, there is a potential distributional shift between the generated solutions and our model's natural output distribution, which can affect the generalization capabilities of the model (Shenfeld et al., 2025; Chu et al.). However, this approach allows us to leverage higher-quality executable solutions that demonstrate successful task completion patterns. The resulting SFT checkpoint serves as the foundation for the subsequent RLVR training.

## 3.2 REINFORCEMENT LEARNING

The reward design is a crucial component of RLVR training. A common choice is to use binary outcome-based rewards for each model response (Luo et al., 2025). For the environment setup task, this means evaluating whether each script successfully configures the corresponding repository. For safety, each script must run in an isolated container, which, together with the massive scale required for efficient RLVR training (e.g., recent work runs up to 512 containers in parallel (Luo et al., 2025)), creates significant computational and technical overhead. To address these challenges, we turn to lightweight execution-free LLM-as-a-Judge reward (denoted $R_{\mathrm{LLM}}$), which serves as a verifiable reward by mimicking rule-based evaluation criteria. The general scheme is presented in Figure 1(b).

To design the reward, we qualitatively study the scripts generated by GPT-4o for a sample of 40 repositories. Overall, we find that failures are due to the inability of the models to fully understand the context of the repository, the system they operate in, and the tools they are required to use. Specifically, we identify 11 failure patterns in model-produced scripts and 3 configuration challenges presented by the repositories that GPT-4o could not overcome. These failures fall into two categories: those producing non-zero exit codes, dominated by incorrect syntax (10% of repositories) and models failing to resolve conflicting dependencies versions (7.5%), and those causing unresolved import issues reported by Pyright, most frequently, models failing to install dependencies present in the codebase but not specified in the configuration files (25%) and optional dependencies required for development, such as test packages or linters (22.5%). A detailed description of the analysis process and all findings are presented in Appendix B.

The reward $R_{\mathrm{LLM}}$ takes in the extracted script $s$ along with a comprehensive context for the corresponding repository and emulates the EnvBench evaluation suite. The instruction for the judge is motivated by our findings of typical errors, and prompts it to predict the exit code from the shell script execution and the number of Pyright issues (`num_issues`). Further implementation details could be found in Appendix A.4. Formally, the reward is calculated as follows:

$$
R_{\mathrm{LLM}}(s) = \begin{cases} -1.0, & \text{if } s \text{ is empty} \\ 0.0, & \text{if } \mathrm{exit\_code}(s) \neq 0 \\ \max\left(1.0 - \frac{\mathrm{num\_issues}(s)}{100}, 0.0\right), & \text{otherwise} \end{cases}
$$

## 4 EXPERIMENTS SETUP

### 4.1 TRAINING SETUP

**Data.** Following recent work on code benchmarks (Gehring et al.; Jain et al., 2025; Le et al., 2022), where agents learn through trial-and-error on the same problems used for evaluation, our setup also employs EnvBench tasks for both training and evaluation. However, we never explicitly provide any ground-truth labels to the model, only rule-based reward scores for the generated scripts. This ensures the model cannot trivially memorize correct answers, forcing it to learn from reward feedback alone. However, to further ensure the absence of memorization, we also (1) reserve 96 repositories as a held-out validation set, using only the remaining 228 repositories for training and (2) evaluate performance on external benchmarks beyond EnvBench. We compare the results on the train and validation sets in Appendix C and find no strong indication of memorization. Due to technical issues, we omit five repositories from EnvBench from our training and validation sets.

**Models.** We select Qwen3-8B as our base model for its strong performance on SE tasks and reasonable compute requirements (Qwen Team, 2025). Qwen models also show consistent improvements with RLVR training compared to other model families (Gandhi et al., 2025). We leave the exploration of other model families and model sizes to future work. We use non-thinking mode because reasoning traces are often long and increase the GPU memory requirements as well as the training duration (Sui et al., 2025).

**Scaffold.** Our experiments follow the zero-shot approach from Eliseeva et al. (2025). The model is prompted with the general task description, predefined context for the particular repository, and information about the base environment (Dockerfile contents). It generates a shell script in a single attempt without receiving any intermediate feedback from the environment. We instruct the model to

provide a script in a Markdown format, enclosed in ```bash and ``` delimiters. The prompts and the provided repository context are described in Appendix A.1.

**SFT Framework and Hyperparameters.**  We employ the LLamaFactory framework (Zheng et al., 2024) using a full-weight training approach. We perform the training with cross-entropy loss on a single H200 GPU for five epochs, without early stopping. We use the AdamW optimizer (Loshchilov and Hutter) and the effective batch size of 16. We reserved 5% of the samples for validation and did not observe signals of overfitting. Comprehensive hyperparameter setup and training details are listed in Appendix A.2.1.

**RL Frameworks and Hyperparameters.**  We use the VeRL framework (Sheng et al., 2024). All our RL training runs are executed on 4xH200 GPUs with all weights optimized by the REIN-FORCE++ (Hu et al., 2025a) algorithm. A more exhaustive comparison with GRPO and GRPO-like objectives is left for future work. We set the batch size of 64 and the number of epochs to 15, yielding 45 training steps. We truncate the prompts longer than 30,000 tokens and allow the model to generate up to 4,096 tokens in response. We use vLLM (Kwon et al., 2023) as the rollout engine and set sampling parameters to the values recommended in the Qwen3 model card for non-thinking mode. We perform 5 optimization epochs on each trajectory batch to improve sample efficiency. We use AdamW (Loshchilov and Hutter) optimizer. We use GPT-4.1 as the backbone LLM for the judge. Comprehensive hyperparameter setup and training details are listed in Appendix A.2.2.

## 4.2 EVALUATION SETUP

**EnvBench-Python (Eliseeva et al., 2025).**  For EnvBench, we extend the original work with three additional metrics. The first one is *pass@5*—the binary measure of success across 5 attempts for each datapoint. It is equal to 1 for a given repository, if at least once in 5 attempts the model was able to generate a script that results in an exit code of 0 and no issues reported by Pyright. Another metric we introduce for more detailed results analysis is *avgFixRate*—the percentage of Pyright issues resolved by running the generated script. To calculate this, we first take the percentage of issues fixed for each repository and then average this number across all repositories. This metric is equal to 100% for the successfully installed repositories, and to 0% for the repositories with a non-zero exit code. We also report *# Failed*—number of repositories where the scripts resulted in a non-zero exit code. All metrics apart from pass@5 are reported averaged over five runs. We use the same base environment, zero-shot scaffold and prompt as during training (Section 4.1), with evaluation infrastructure available in our replication package.

**Repo2Run (Hu et al., 2025b).**  For Repo2Run, we also use the *pass@5* metric. Success is determined by running test collection via pytest: if there are no collection errors, the setup is considered successful. We do not use the agentic setting from original work and instead employ the same base environment as the EnvBench.

**Terminal-Bench (The Terminal-Bench Team, 2025).**  For Terminal-Bench, which is a more challenging benchmark, especially for smaller models, we use the *pass@10* metric. The benchmark provides custom evaluation commands for each data point.

**Baselines.**  We compare the trained models against multiple general-purpose LLMs. We evaluate three closed-source OpenAI models: GPT-5, GPT-4o, and GPT-4o-mini. We also assess multiple models from the Qwen3 family (8B, 14B, and 32B parameters) to understand how our approach compares across different model scales. All Qwen3 models are evaluated in non-thinking mode for consistency.

## 5 RESULTS

### 5.1 TRAINING DYNAMICS

Training dynamics of base and SFT model with the LLM-based reward described in Section 3.2 are depicted in Figure 2. The reward function returns values from the $[-1, 1]$ range, where $-1$ indicates malformatted scripts, and 1 indicates perfect performance.

Table 1: Results on Repo2Run and Terminal-Bench for base models and our tuned Qwen3-8B. For Repo2Run, success is determined as a zero exit code and no test collection errors. For Terminal-Bench, success is determined by per-sample evaluation commands. Our PIPER model achieves the best performance on Repo2Run. However, SFT-based models underperform on Terminal-Bench's multi-turn setting.

| Model | Repo2Run | Terminal-Bench |
|---|---|---|
| | pass@5 | pass@10 |
| GPT-5 | ❶ 106 | ❶ 45 |
| GPT-4o | 67 | ❷ 25 |
| GPT-4o-mini | 84 | 19 |
| Qwen3-32B | 71 | ❸ 23 |
| Qwen3-14B | 64 | 14 |
| Qwen3-8B | 32 | 8 |
| PIPER | ❷ 103 | 4 |
| PIPER$^{\text{RL-only}}$ | 77 | 9 |
| PIPER$^{\text{SFT-only}}$ | ❸ 98 | 2 |

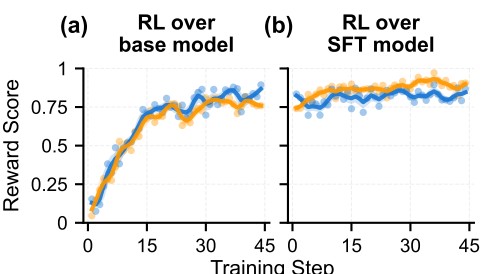

Figure 2: RLVR training dynamics with the proxy rewards described in Section 3.2. Raw datapoints are shown as semi-transparent dots, with Gaussian-smoothed curves overlaid to highlight trends. **Blue** shows average reward on the training set; **orange** shows average reward on the validation set. The x-axis is training steps, and the y-axis is average reward. Evolution of the LLM-as-a-Judge reward $R_{\text{LLM}}$ **(a)** over the base model, **(b)** over the SFT model.

The base model exhibits formatting compliance but fails to satisfy the meaningful criteria imposed by the reward. This is evident from almost zero **validation** scores at step 0, which are close to the minimum values achievable with the correct formatting. On the other hand, the SFT checkpoint from the start produces high-quality scripts that are correctly formatted and highly assessed by the judge.

We observe a steady initial increase for both **training** and **validation** sets, which then slows for the base model, and plateaus for the SFT model. The substantial differences in validation reward scores between step 0 and step 45 suggest that RLVR training successfully steers the models to better adhere to the criteria imposed by the reward. In addition, we do not observe strong overfitting: there is only a small gap between **training** and **validation** reward scores.

## 5.2 EVALUATION RESULTS

The evaluation results on EnvBench are presented in Table 2. While the GPT-5 frontier model claims the first place, the proposed PIPER model is competitive with both strong open-source (Qwen3-32B) and closed-source (GPT-4o) baselines. With respect to all reported metrics on EnvBench, it comes with a small gap or is on par with these strong competitors. The outlier metric here is the surprisingly low *# Failed* of the GPT-4o-mini model, which claims the second place in the rating, and is significantly better than GPT-4o. We leave a thorough investigation of this phenomenon to future work.

To assess the cost-performance ratio of PIPER, we compare the baselines with respect to both performance and inference cost. We take prices per 1M generated tokens as the cost of a model. For OpenAI models, we take official API prices[5], and for the Qwen3 model family, we take costs from the Alibaba Cloud website[6]. The costs are reported as of 22.09.2025. Figure 3(b) indicates comparable-or-better performance relative to baselines at a fraction of their cost; PIPER can also run on local machines, further reducing cost.

The ablation of RL and SFT phases, also shown in Table 2, shows the necessity of the two-stage pipeline. While both SFT and RL checkpoints outperform the base model, pushing its *avg@5* performance from 2.6 to 13 (SFT) or 11.8 (RL), they both are significantly worse than the checkpoint yielded by the combined training.

---

[5]https://platform.openai.com/docs/pricing
[6]https://www.alibabacloud.com/help/en/model-studio/models

Table 2: EnvBench evaluation results for base models and PIPER with various training setups. The total number of samples is 329. **pass@5** shows the number of successful samples (zero exit code and zero issues). **avg@5** shows mean $\pm$ std for the following metrics: **# Success** (average number of successful samples per run), **# Failed** (average number of samples where scripts finished with non-zero exit code), and **avgFixRate** (average ratio of resolved import issues per sample as compared to the evaluation run with empty setup script; for samples where scripts execute with non-zero exit codes, ratio is considered 0). The symbol $\uparrow$ indicates higher is better, while $\downarrow$ indicates lower is better.

| Model | pass@5 | avg@5 | | |
|---|---|---|---|---|
| | # Success $\uparrow$ | # Success $\uparrow$ | # Failed $\downarrow$ | avgFixRate $\uparrow$ |
| GPT-5 | ❶ 43 | ❶ $25 \pm 3$ | ❶ $131 \pm 5$ | ❶ $(32.7 \pm 3.3)\%$ |
| GPT-4o | ❷ 29 | ❷ $19 \pm 2$ | $194 \pm 6$ | ❷ $(28.0 \pm 1.0)\%$ |
| GPT-4o-mini | 15 | $9.6 \pm 1.3$ | ❷ $166 \pm 6$ | $(22.6 \pm 1.5)\%$ |
| Qwen3-32B | ❷ 29 | ❸ $16.2 \pm 1.3$ | $207 \pm 6$ | $(25.1 \pm 1.3)\%$ |
| Qwen3-14B | 17 | $5.6 \pm 1.1$ | $268 \pm 10$ | $(9.95 \pm 0.81)\%$ |
| Qwen3-8B | 8 | $2.6 \pm 1.5$ | $294 \pm 2$ | $(4.4 \pm 1.2)\%$ |
| PIPER | ❸ 27 | ❷ $19 \pm 3$ | ❸ $183 \pm 3$ | ❸ $(27.2 \pm 1.2)\%$ |
| PIPER$^{\text{SFT-only}}$ | 25 | $13.0 \pm 1.0$ | $192 \pm 7$ | $(23.6 \pm 1.4)\%$ |
| PIPER$^{\text{RL-only}}$ | 19 | $11.8 \pm 0.8$ | $205 \pm 5$ | $(25.2 \pm 1.2)\%$ |

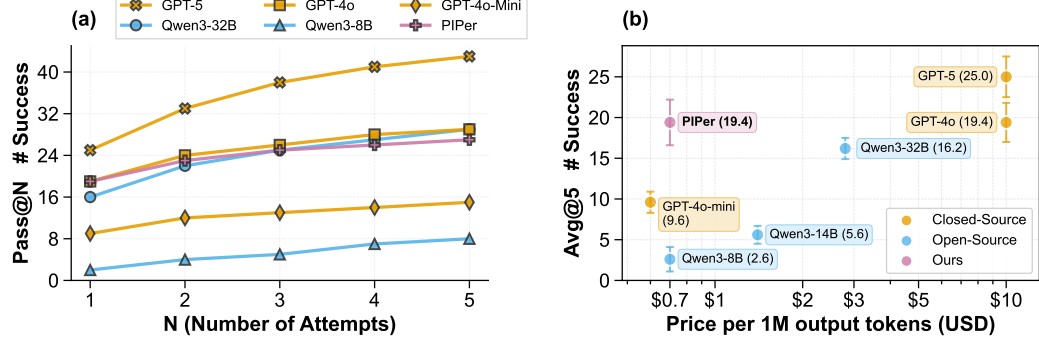

Figure 3: Performance analysis of environment setup models on EnvBench-Python. **(a)** Pass@$N$ performance showing how model success rates improve with multiple attempts ($N = 1$ to $5$). Our PIPER model (shown with cross markers) achieves performance comparable to much larger models like GPT-4o and Qwen3-32B, while substantially outperforming the base Qwen3-8B model. **(b)** Cost-performance tradeoff analysis comparing average *pass@1* performance (averaged over five runs) against price per 1M output tokens (USD).

Finally, in the Figure 3(a), we explore how the model results improve with multiple attempts. While the scaling of PIPER is slower than that of the strong baselines, it is still able to beat them at cost parity. For example, *pass@3* of the proposed model is higher than the *pass@2* of both GPT-4o and Qwen3-32B models. Also, *pass@5* of PIPER (27) is higher than *pass@1* of GPT-5 (25), while the cost of inference is more than 14 times lower.

## 5.3 GENERALIZATION

To ensure that the results of the evaluation are not based on overfitting, we separately evaluate our model on the evaluation subset of EnvBench and on two additional datasets. We detail the scores on

the evaluation subset of EnvBench in Appendix C, but notice that the model yields the results better or on par with the strong baselines. This confirms that there are no strong signs of memorization.

Table 1 shows the evaluation of the PIPER on Repo2Run and Terminal-Bench. On Repo2Run, which shares similar single-turn Python environment setup characteristics with EnvBench, all our trained models substantially outperform the base Qwen3-8B (32 success cases), with PIPER achieving the best results (103 success cases) and even surpassing larger models like Qwen3-32B (71 success cases) and GPT-4o-mini. However, on Terminal-Bench, which requires multi-turn agentic interactions for system configuration tasks, we observe a different pattern: while the PIPER$^{\text{RL-only}}$ model shows modest improvement (8 to 9 success cases), PIPER$^{\text{SFT-only}}$ (2) actually underperforms the base model, with united training procedure of PIPER showing a slight recovery (4). This suggests that while SFT improves single-turn performance, it struggles with the multi-turn interactions required by Terminal-Bench. These cross-benchmark results demonstrate that our proxy reward-based RLVR training develops transferable shell scripting capabilities, with the RL component providing more robust generalization across diverse interaction paradigms than supervised fine-tuning alone.

## 6 RELATED WORK

**Environment Setup.** Following the advances of LLMs in other SE tasks (Liu et al., 2024), previous works extensively explored their applications to the environment setup task. Several environment setup benchmarks were introduced, such as EnvBench (Eliseeva et al., 2025), Repo2Run (Hu et al., 2025b), and others (Milliken et al., 2025; Arora et al., 2025). They differ in scale (from tens to hundreds of repositories), expected model outputs (shell scripts or Dockerfiles), and metrics (static analysis or test-based). Our study required a large sample of Python repositories, which left us with EnvBench (329 repositories) and Repo2Run (420 repositories). We selected EnvBench for training because its construction process explicitly prioritizes challenging repositories, providing a diverse learning signal.

Existing environment setup approaches range from simple zero-shot prompts (Badertdinov et al., 2025; Eliseeva et al., 2025; Li et al., 2025) to complicated agentic workflows (Milliken et al., 2025; Bouzenia and Pradel, 2025; Hu et al., 2025b; Vergopoulos et al., 2025; Zhang et al., 2025; Guo et al., 2025). Existing works use general-purpose LLMs as backbones, and many workflows include execution of intermediate agent outputs (Eliseeva et al., 2025; Milliken et al., 2025; Bouzenia and Pradel, 2025; Hu et al., 2025b; Vergopoulos et al., 2025; Zhang et al., 2025; Guo et al., 2025), introducing isolation and cost considerations. In contrast, we focus on a zero-shot scaffold, which was previously shown to achieve reasonable performance given its simplicity (Eliseeva et al., 2025; Badertdinov et al., 2025), to study how far LLMs can go under consistent constraints. Finally, we note that many works use automated environment setup approaches as a mere tool for constructing SWE-bench-like (Jimenez et al., 2024) datasets (Badertdinov et al., 2025; Vergopoulos et al., 2025; Zhang et al., 2025; Guo et al., 2025), making the environment setup not the primary research focus.

**Reinforcement Learning with Verifiable Rewards (RLVR).** Reinforcement Learning (RL) has emerged as a powerful LLM post-training technique to further enhance the model's capabilities, with early successes achieved from human feedback (Christiano et al., 2017; Kaufmann et al., 2024). Building on this foundation, the RLVR has gained traction, wherein the reward signal is provided by a rule-based or programmatic verifier. RLVR has found particularly impactful applications in domains such as mathematics (Lambert et al., 2025; Feng et al., 2025) and code generation (Wei et al., 2025; Luo et al., 2025; Golubev et al., 2025).

The effectiveness of RLVR has been amplified by recent advances in RL algorithms building upon Proximal Policy Optimization (PPO) (Schulman et al., 2017) (e.g., VAPO (Yue et al., 2025), RLOO (Kool et al., 2019; Ahmadian et al., 2024), Reinforce++ (Hu et al., 2025a), GRPO (Shao et al., 2024), DAPO (Yu et al., 2025), Dr. GRPO (Liu et al., 2025a), GRPO++ (Luo et al., 2025), GSPO (Zheng et al., 2025)). Furthermore, recent research has explored RLVR settings that do not rely on labeled data (Zhao et al., 2025) or even operate without an explicit verifier (Zhou et al., 2025). Recent comparative studies of RL and SFT training approaches and their combinations have revealed both synergistic improvements (Liu et al., 2025b; Yoshihara et al., 2025) and potential degradation of final model performance (Chen et al., 2025a), while also demonstrating that SFT alignment can impair models' generalization capabilities (Shenfeld et al., 2025; Liu et al., 2025b; Wu et al., 2025).

## 7 LIMITATIONS AND FUTURE WORK

**Models** We apply the proposed framework to a single LLM, Qwen3-8B in non-thinking mode. While it comes from the widely used Qwen3 family and presents a competitive quality-compute tradeoff, the range of applicability of our study could be further verified by probing other model families, different model sizes, and reasoning LLMs.

**Scaffold** We consider a simple single-turn scaffold in our experiments. Previous works on environment setup suggest that multi-turn agentic scaffolds—which iteratively interact with an environment and refine their solutions based on the feedback received on each step—could bring significant improvements. Extending RLVR training to such multi-turn scaffolds represents a natural progression for enhancing environment setup capabilities.

**Proxy Rewards** We introduce the lightweight LLM-based reward function that allows for the RLVR training pipeline without computational overhead on scaling containerized execution. While we consider this direction promising given its light computation burden and obtained results, ground truth runtime feedback would likely provide richer training signals and drive further performance gains.

## 8 CONCLUSION

We presented PIPER—a strong on-device-sized model for environment setup. It is trained with a two-stage pipeline without ground truth data, with SFT distillation to teach the model to write correct scripts, and RLVR to further improve the environment setup capabilities. We use a lightweight reward that mimics a ground truth execution check with the LLM-as-a-Judge technique to lift strong infrastructure requirements for the direct environment feedback. The resulting model performs on par or better than several times more expensive models, such as GPT-4o and Qwen3-32B. Importantly, our findings extend beyond environment setup. The trained models maintain reasonable performance on the out-of-distribution Terminal-Bench in an agentic scaffold, indicating genuine improvement of terminal manipulation capabilities rather than task-specific overfitting. Our replication package with training code and trained model checkpoints is available online: `https://github.com/PIPer-iclr/PIPer`.

## 9 REPRODUCIBILITY STATEMENT

To ensure full reproducibility of our results, all model checkpoints mentioned in this paper are publicly available, including PIPER[RL-only], PIPER[SFT-only], and the final PIPER model, along with all raw evaluation results from our experiments (`https://huggingface.co/PIPer-iclr`). The complete codebase used for SFT training, RL training, and evaluation is available in our dedicated repository (`https://github.com/PIPer-iclr/PIPer`), which contains all configuration files and implementation details. Upon acceptance of this work, we will also publish the complete training run logs to provide full transparency into the training process and enable detailed analysis of our experimental procedures.

## 10 ACKNOWLEDGEMENTS

We acknowledge the use of LLMs for text polishing and minor language improvements throughout this manuscript. All technical content, ideas, and substantial writing remain the original work of the authors.

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

## A  IMPLEMENTATION DETAILS

In this section, we provide additional details on our experiments.

### A.1  SCAFFOLD DETAILS

We use the same zero-shot scaffold as in Eliseeva et al. (2025). The prompt is provided in Figure 4. We collect the repository context by running the following bash commands:

```
tree -a -L 3 --filelimit 100 || ls -R
for f in README.md INSTALL.md SETUP.md docs/INSTALL.md docs/SETUP.md; do
  if [ -f "$f" ]; then echo -e "\n=== $f ==="; cat "$f"; fi
done
find . -type f \( \
  -name "*requirements*.txt" -o -name "setup.py" -o -name "pyproject.toml" -o -name "
      setup.cfg" -o -name "tox.ini" \
\) | while read f; do echo -e "\n=== $f ==="; cat "$f"; done
find . -type f -name "*.py" -exec grep -l "python_version\|python_requires" {} \;
find . -type f \( -name ".env*" -o -name "*.env" -o -name "Dockerfile*" \) | \
while read f; do echo -e "\n=== $f ==="; cat "$f"; done
```

### A.2  TRAINING DETAILS

#### A.2.1  SFT TRAINING

We show the hyperparameters used in our SFT training in Table 3. We fine-tune the Qwen3-8B model for 5 epochs with a learning rate of $5 \times 10^{-5}$ using the AdamW optimizer with cosine scheduling, weight decay of $0.01$, gradient accumulation of 4 steps, and batch size of 4. Scripts, selected for the training, cover $227/228$ unique repositories from the train split with a median sample size of 11 for each repository. Training is performed with bfloat16 precision, with a $5\%$ validation split for evaluation.

756
757
758
759
760
761
762
763
764
765
766
767
768
769
770
771
772
773
774
775
776
777
778
779
780
781
782
783
784
785
786
787
788
789
790
791
792
793
794
795
796
797
798
799
800
801
802
803
804
805
806
807
808
809

**Zero-shot Prompt Overview**

**System Message:**
Your task is to generate a `bash` script that will set up a Python development environment for a repository mounted in the current directory.
You will be provided with repository context. Follow the build instructions to generate the script.

A very universal script might look like this:

    {baseline_script}

However, your job is to make a script more tailored to the repository context.
It will be only run on a single repository mounted in the current directory that you have information about.
The script must not be universal but setup the environment just for this repository.
Avoid using universal if-else statements and try to make the script as specific as possible.

The script should:

- Install the correct Python version based on repository requirements
- Install all project dependencies from `requirements.txt`, `setup.py`, or `pyproject.toml`
- Install any required system packages

For reference, the script will run in this Docker environment, so most of the tools you need will be available:

    {dockerfile}

IMPORTANT:

- Generate ONLY a `bash` script – you cannot interact with the system
- The script must be non-interactive (use `-y` flags where needed)
- Base all decisions on the provided repository context. Follow the context instructions.
- Do not use `sudo` – the script will run as root
- If you use `pyenv install`, please use the `-f` flag to force the installation. For example: `pyenv install -f $PYTHON_VERSION`
- The script must be enclosed in ```` ```bash ``` ```` code blocks

**User Message:**
Repository Context:
`context`
Generate a complete `bash` script that will set up this Python environment.
The script must be enclosed in ```` ```bash ``` ```` code blocks, it can rely on the tools available in the Docker environment.

Figure 4: Prompt for the zero-shot scaffold for the environment setup task from Eliseeva et al. (2025). Baseline script and Dockerfile context variables are the same as theirs. Repository context is collected by executing a fixed set of commands within the repository in the target Docker environment.

Table 4: RL parameters for VeRL

| Parameter | Value |
|---|---|
| **Model Configuration** | |
| Max Prompt Length | 30,000 |
| Max Response Length | 4,096 |
| **Training Settings** | |
| Train Batch Size | 64 |
| Mini-Batch Size | 32 |
| Micro-Batch Size | 1 |
| Optimizer | AdamW |
| Learning Rate | 5e-6 |
| Gradient Clipping | 1.0 |
| Total Steps | 45 |
| **RL Settings** | |
| Algorithm | Reinforce++ |
| KL Loss | False |
| KL Reward | False |
| Entropy Coefficient | 0.001 |
| PPO Epochs | 5 |
| $N$ Rollouts | 1 |
| Rollout Temperature | 0.7 |
| Rollout Top-P | 0.8 |
| Rollout Top-K | 20 |

Table 3: SFT parameters for LLaMA-Factory

| Parameter | Value |
|---|---|
| **Training Settings** | |
| Epochs | 5 |
| Learning Rate | 5e-5 |
| Weight Decay | 0.01 |
| Optimizer | AdamW |
| LR Scheduler | Cosine |
| Gradient Accumulation Steps | 4 |
| Warmup Ratio | 0.1 |
| Max Grad Norm | 1.0 |
| Batch Size | 4 |
| **Precision & Optimization** | |
| Dtype | bfloat16 |
| FlashAttention-2 | enabled |
| **Evaluation & Logging** | |
| Validation Split | 0.05 |
| Early Stopping | — |

### A.2.2 RL TRAINING

We show the hyperparameters used in our RL training in Table 4. Sampling parameters are set to the values recommended in the Qwen3 model card[7] for non-thinking mode. Full configuration files and code are available in the reproduction package.

### A.3 EVALUATION DETAILS

**EnvBench.** We build off the original implementation provided by EnvBench authors. For Qwen3 models, we set the sampling parameters to the values recommended in the corresponding model cards, same as for training (Appendix A.2). The resulting evaluation suite is available in our replication package.

**Repo2Run.** As Repo2Run replication package only includes code for inference of the proposed Repo2Run agent, we extend EnvBench evaluation suite to support repositories and success check (test collection via pytest) from Repo2Run. We use the same zero-shot scaffold and prompts as for EnvBench, detailed in Appendix A.1. The resulting evaluation suite is available in our replication package.

**Terminal-Bench.** We use the original implementation to run the evaluation on Terminal-Bench. We use Terminus 1 scaffold and version 0.1.1 of the benchmark.

### A.4 LLM-AS-A-JUDGE REWARD IMPLEMENTATION

The LLM-as-a-Judge reward provides repository-specific, scalable feedback for environment setup scripts by using an LLM as an evaluator. The LLM is prompted to simulate the execution of a candidate shell script in EnvBench Docker environment and predict the outcome of the environment setup process, including the script's exit code and the number of missing import issues (as would be detected by Pyright static analysis).

The prompt provided to the LLM includes the following components: the Dockerfile specifying the environment, evaluation guidelines informed by our exploratory analysis of model-generated scripts,

---

[7]https://huggingface.co/Qwen/Qwen3-8B#best-practices

and several few-shot examples illustrating script grading. Complete prompt templates and reward implementation code are available in our replication package.

We selected GPT-4.1 as the language model for our experiments, as it consistently yielded the most reliable results. While we also evaluated GPT-4o and GPT-4o-mini, these models did not achieve comparable performance. In addition, we explored several ablations: (1) augmenting the LLM-as-a-Judge with repository information like the zero-shot context, and (2) replacing the LLM-as-a-Judge with an LLM Agent equipped with tools for repository exploration. Neither approach led to a noticeable improvement in model performance. Consequently, we adopted the simplest and most robust configuration for our main experiments.

## B  EMPIRICAL STUDY OF ENVIRONMENT SETUP FAILURE PATTERNS

We manually analyzed scripts generated by GPT-4o in a zero-shot scaffold, the second-best approach on EnvBench, to understand the fault modes of environment setup scripts. Specifically, we selected 40 scripts (from the first 40 repositories in lexicographical order where the results were available). Out of those repositories, 2 were set up correctly, 16 had a non-zero exit code (failed), and 22 had unresolved import issues. For each script, we collected free-form observations about potential failure reasons and applied an open coding approach to extract common failure themes.

We present the resulting failure patterns in Table 5, with labels for 40 repositories available in our replication package. We identify three failure patterns categories: (i) Script Problems, the explicit mistakes made in the model-generated scripts, (ii) Repository Problems, the configuration challenges presented by a specific repository that the model failed to consider, and (iii) Eval Problems, runtime failures of EnvBench evaluation suite and/or limitations of the static analysis. Most failures are caused by Script Problems, while unresolved import issues are often due to Repository Problems. We observe 3 Eval Problems in total (7.5% of 40 repositories sample).

## C  TRAIN/VALIDATION PERFORMANCE

To rule out memorization from improvements of our models that were trained on a part of EnvBench, we present experimental results separately on the held-out validation set in Table 6. From Table 6, we observe that all PIPER variants retain substantial improvements over the base Qwen3-8B on the validation set. Similarly, the performance of PIPER on the validation set is comparable to Qwen3-32B and GPT-4o, with either second-best or third-best results across all the considered metrics. GPT-5 remains the strongest among the considered baselines. Mirroring our findings on full EnvBench, SFT-only and RL-only checkpoints show lower per-run metrics than PIPER that combines both stages.

Table 5: Identified environment setup failure patterns for zero-shot GPT-4o for 40 repositories (percentages are relative to full 40 repositories sample). **# Failure** means number of failed repositories which contain given pattern; **# Issues** — number of repositories with unresolved import issues. Note that each repository can contain multiple fault patterns.

| Failure Pattern | Explanation | # Failure | # Issues |
|---|---|---|---|
| **Script Problems** | | | |
| Wrong Syntax | Syntax errors in the script. | 4 (10%) | — |
| Dependencies Resolution Issue | Dependency manager can't resolve dependencies due to conflicting versions. | 3 (7.5%) | — |
| Multiple Dep. Managers | Script uses both pip and Poetry. | 2 (5%) | — |
| Wrong Python Binary | Script installs dependencies for a specific Python binary, but fails to configure the system to use that binary. | 2 (5%) | 1 (2.5%) |
| Missing System Package | Script doesn't install a system package required by repository dependencies. | 1 (2.5%) | — |
| Non-existent Package | Script tries to install a package that does not exist on PyPI. | 1 (2.5%) | — |
| Wrong Operation | Script executes a command that conflicts with the given base environment (e.g., tries to install Poetry even though it is already installed). | 3 (7.5%) | — |
| Wrong Python Version | Script uses Python version conflicting with repository requirements. | 1 (2.5%) | 1 (2.5%) |
| Missing Dep. Group | Script does not install an optional dependency group required for development (e.g., `test`). | — | 9 (22.5%) |
| No Editable Mode | Script installs the repository in non-editable mode not suitable for development (relevant for pip). | — | 3 (7.5%) |
| Missing Configuration File | Script does not install dependencies from a configuration file in the repository (e.g., multiple `requirements-dev`, `requirements-docs`, etc.). | — | 2 (5%) |
| **Repository Problems** | | | |
| Requirements Not Specified | Some packages used in the repository codebase are not specified in the configuration files. | — | 10 (45.5%) |
| Poetry Lock Outdated | `poetry install` fails because the `poetry.lock` file must be regenerated first. | 2 (12.5%) | — |
| Misconfigured PYTHONPATH | Local modules do not resolve correctly because the `PYTHONPATH` environment variable is not configured properly. | —- | 2 (9.09%) |
| **Eval Problems** | | | |
| Dynamic Imports | Repository includes dynamic imports that cannot be resolved with static analysis. | — | 5 (12.5%) |
| Eval Failure | Runtime failure of EnvBench evaluation suite, not associated with specific script or repository characteristics. | 2 (12.5%) | — |
| Hardware Problems | Dependencies require hardware not available in the base environment (e.g., GPU). | 1 (6.25%) | — |

Table 6: **Validation** split results for base models and PIPER model variations. Total number of samples is 96. **pass@5** shows the number of successful samples (zero exit code and zero issues). **avg@5** shows mean $\pm$ std for the following metrics: **# Success** (average number of successful samples per run), **# Failed** (average number of samples where scripts finished with non-zero exit code), and **avgFixRate** (average ratio of resolved import issues per sample as compared to the evaluation run with empty setup script; for samples where scripts execute with non-zero exit codes, ratio is considered 0). The symbol $\uparrow$ indicates higher is better, while $\downarrow$ indicates lower is better.

| Model | pass@5 | avg@5 | | |
|---|---|---|---|---|
| | # Success $\uparrow$ | # Success $\uparrow$ | # Failed $\downarrow$ | avgFixRate $\uparrow$ |
| GPT-5 | ❶ 10 | ❶ $7.4 \pm 1.5$ | ❶ $36.0 \pm 3.1$ | ❶ $(41.4 \pm 2.8)\%$ |
| GPT-4o | ❸ 6 | ❸ $4.8 \pm 1.1$ | $60.4 \pm 1.5$ | ❸ $(26.5 \pm 1.1)\%$ |
| GPT-4o-mini | 3 | $2.0 \pm 0.7$ | ❷ $50.8 \pm 2.0$ | $(20.4 \pm 0.8)\%$ |
| Qwen3-32B | ❷ 7 | ❸ $4.8 \pm 0.4$ | $62.6 \pm 2.3$ | $(25.3 \pm 1.4)\%$ |
| Qwen3-14B | 4 | $1.8 \pm 0.4$ | $81.6 \pm 1.7$ | $(10.4 \pm 1.5)\%$ |
| Qwen3-8B | 1 | $0.2 \pm 0.4$ | $89.6 \pm 1.1$ | $(3.2 \pm 1.9)\%$ |
| PIPER | ❸ 6 | ❷ $5.2 \pm 0.8$ | ❸ $54.2 \pm 1.6$ | ❷ $(30.0 \pm 1.9)\%$ |
| PIPER$^{\text{SFT-only}}$ | ❸ 6 | $3.2 \pm 0.4$ | $57.2 \pm 1.3$ | $(24.6 \pm 2.3)\%$ |
| PIPER$^{\text{RL-only}}$ | ❸ 6 | $3.8 \pm 0.8$ | $63.0 \pm 3.1$ | $(24.6 \pm 1.9)\%$ |

