# OpenReview forum: "PIPer: On-Device Environment Setup via Online Reinforcement Learning"
_ICLR.cc/2026/Conference — Submitted to ICLR 2026_

### Official Review · Reviewer_F4KY · 2025-10-28

**Soundness:** 3
**Presentation:** 3
**Contribution:** 2
**Rating:** 4
**Confidence:** 4

**Summary:**

This paper tackles repository environment setup by training a small open model with a two-stage recipe: supervised distillation from a stronger teacher to learn executable setup scripts, followed by online reinforcement learning with a lightweight, verifiable reward (LLM-as-a-judge predicting exit status and import issues). The goal is to close the gap between an 8B on-device model and larger closed/open foundation models under a single-turn “generate one script” protocol. The method is evaluated on EnvBench-Python, Repo2Run, and Terminal-Bench, reporting improved pass@N over same-size baselines and competitive scores relative to much larger models at substantially lower cost.

**Strengths:**

1) Practical problem with real engineering impact: automating per-repo environment setup is valuable for reproducibility, onboarding, and CI health.
2) Clear training recipe: SFT establishes executable script priors; online RL refines toward pass/failure-oriented behaviors with a cheap reward signal.
3) Sensible evaluation across multiple benchmarks (EnvBench-Python, Repo2Run), with pass@N reporting that reflects realistic multi-try usage.
4) Cost/efficiency perspective: shows that a tuned 8B model can approach larger models’ performance in the single-turn setting with better inference cost.

**Weaknesses:**

1) Only one base model (Qwen3-8B) is trained, which is not comprehensive enough.
2) The baselines are primarily foundation models; this does not sufficiently demonstrate the method’s effectiveness compared to task-specific systems.
3) Reporting under-emphasizes the common single-try metric: pass@1 is not tabulated across benchmarks. The paper includes a figure that visualizes an average pass@1 comparison but does not provide per-model pass@1 tables—and there is no explicit textual explanation for omitting tabulated pass@1—reducing comparability to standard single-attempt settings.

**Questions:**

1) Could you include results for a second open base model (e.g., another 7–9B family) to test robustness of the training recipe?
2) Can you add a task-specific baseline (e.g., a standardized agentic scaffold with controlled tools/budget) to better isolate method gains vs. model size?
3) Please provide per-repository runtime/cost breakdowns and scaling with repo size to support deployment decisions.

---

> ### Author Response · Authors · 2025-11-20
> **Response to reviewer F4KY**
>
> > Could you include results for a second open base model (e.g., another 7–9B family) to test robustness of the training recipe?
> >
>
> **Response:**
> We agree with this request, and are performing the additional ablation studies. We expect to report on the results within a couple of days.
>
> > Can you add a task-specific baseline (e.g., a standardized agentic scaffold with controlled tools/budget) to better isolate method gains vs. model size?
> >
>
> **Response:**
> We investigated the performance of several baselines in the Bash Agent scaffold [1] tailored for the environment setup task. It follows a widely used ReAct paradigm [2], providing the model with access to a Bash terminal with which it can interact iteratively. We use the original code and hyperparameters, allowing the models up to 30 iterations.
>
> We will include the full results in the camera-ready version, but we would like to note, that while this agentic approach improves the baseline model (Qwen3-8B) from 8 to 14 (pass@5) and from 2.6 to 4.8 (pass@1), PIPer still outperforms it on both metrics (27 pass@5 and 19 pass@1), while being cheaper due to the iterative nature of the agent. The agentic setup notably boosts the GPT-4o model on the pass@5 metric (from 29 to 36), but lowers the pass@1 (from 19 to 17.4). As a result, while this work was focused on the single-step environment setup, we consider it is competitive with the agentic setup applied to the models of the same scale. We acknowledge the limitations of the current approach and the future line of work in Section 7.
>
> > Please provide per-repository runtime/cost breakdowns and scaling with repo size to support deployment decisions.
> >
>
> **Response:**
>
> We acknowledge the importance of performance statistics for practitioners and will add statistics on the consumed tokens to the appendix. Across all our evaluation runs on EnvBench, prompts require 7k tokens per repository on average, while model completions require 200 tokens, resulting in a total token consumption of 7.2k. On average, evaluation takes 355 seconds per repository. We observe a moderate Spearman correlation between the number of files in the repository and the total number of tokens for model inference (0.56), but a negligible correlation for runtime (-0.01). While correlation for the number of tokens suggests scaling with repository size, even for the largest repository we have, with 31k files, the total number of tokens is 8.8k. We hypothesize that the number and size of repository dependencies are the primary factors influencing runtime.
>
> > Reporting under-emphasizes the common single-try metric: pass@1 is not tabulated across benchmarks. The paper includes a figure that visualizes an average pass@1 comparison but does not provide per-model pass@1 tables—and there is no explicit textual explanation for omitting tabulated pass@1—reducing comparability to standard single-attempt settings.
> >
>
> **Response:**
> We apologize for any confusion in our presentation, and will clarify this in the camera-ready version. The pass@1 EnvBench results are reported in Table 2 under the "avg@5 # Success" column, which shows the average number of successful setups across five independent runs. For the two additional benchmarks (Repo2Run and Terminal-Bench), we report pass@1 results here and will add them to the paper:
>
> | Model | Repo2Run | Terminal-Bench |
> |-------|----------|----------------|
> |  | avg pass@1| pass@1 |
> | GPT-5 | 41.2 ± 22.4 | 20 |
> | GPT-4o | 32.0 ± 10.7 | 9 |
> | GPT-4o-mini | 54.6 ± 2.9 | 9 |
> | Qwen3-32B | 41.6 ± 4.3 | 9 |
> | Qwen3-14B | 31.2 ± 4.7 | 5 |
> | Qwen3-8B | 9.6 ± 2.5| 3 |
> | PIPer | 79.2 ± 4.4| 4 |
> | PIPer-RL-only | 52.0 ± 4.6 | 4 |
> | PIPer-SFT-only | 62.0 ± 2.3 | 1 |
>
> *NB! We report Terminal-Bench pass@1 results from the original evaluation suite. In the camera-ready version of the paper, we will include average values.*
>
> Additionally, for enhanced clarity, we will  include in the supplementary materials the complete per-model, per-repository success matrices for EnvBench [3] and a visualization comparing repository-level success patterns across models [4]
>
> [1] Eliseeva, A., Kovrigin, A., Kholkin, I., Bogomolov, E., & Zharov, Y. EnvBench: A Benchmark for Automated Environment Setup. In ICLR 2025 Third Workshop on Deep Learning for Code.
>
> [2] Yao, S., Zhao, J., Yu, D., Du, N., Shafran, I., Narasimhan, K. R., & Cao, Y. (2022, October). ReAct: Synergizing Reasoning and Acting in Language Models. In The eleventh International Conference on Learning Representations.
>
> [3] https://github.com/PIPer-iclr/misc/blob/main/piper_envbench_results.md
>
> [4] https://github.com/PIPer-iclr/misc/blob/main/model_performance_per_datapoint.png?raw=true

---

> ### Author Response · Authors · 2025-11-26
> **Response to reviewer F4KY (follow-up)**
>
> > Could you include results for a second open base model (e.g., another 7–9B family) to test robustness of the training recipe?
> >
>
> **Response:** We sincerely thank the reviewer for their patience while we conducted additional training and evaluations. To thoroughly address this question and demonstrate the robustness of the proposed method, we have deployed it across multiple model families and sizes, focusing on the small, on-device-sized models. Specifically, we experimented with Qwen3-4B [1], Mistral-7B [2], Ministral-8B [3], Llama-3-8B-Instruct [4], and Llama-3.1-8B-Instruct [5]. We applied the same hyperparameters reported in the paper across all models.
>
> We attach the resulting table in this comment and will use the 1-page extension of the paper to report and discuss this ablation study. All new trained model checkpoints, along with their associated training dynamics plots, have been published on our HuggingFace [6] and GitHub repository [7] for full transparency and reproducibility.
>
> Due to the limited resources, we haven’t tuned the hyperparameters of our method for each of the models under investigation, and we acknowledge that the results of each individual model can be further improved. However, across all model families and sizes, we observe behaviour similar to that of the Qwen3-8B presented in the paper. The two models that stand out are Qwen3-4B and Llama-3-8B-Instruct. Neither of them improved during the RL-only phase, but both show improvement during the SFT+RL phase. For the Qwen3-4B, we hypothesize that this may be due to the base model's inability to generate well-formatted scripts, thereby hindering RL training. And for Llama-3-8B-Instruct, it is a known limitation [8, 9, 10]. Driven by the results of various model sizes, we hypothesize that one important hyperparameter of the proposed method is the balance between SFT and RL. While the smaller models may require more SFT examples, the larger ones may be impeded by this phase. We leave a thorough research of this dependency for further work.
>
> | **Model** | **pass@5 #Success ↑** | **avg@5 #Success ↑** | **avg@5 #Failed ↓** | **avg@5 avgFixRate ↑** |
> | --- | --- | --- | --- | --- |
> | Mistral-7B (SFT + RL) | 22 | 16.6 ± 1.5 | 169.8 ± 4.2 | (29.42 ± 0.71)% |
> | Mistral-7B (SFT-only) | 18 | 12.4 ± 1.7 | 211.8 ± 5.8 | (21.56 ± 0.83)% |
> | Mistral-7B (RL-only) | 8 | 3.4 ± 0.5 | 181.6 ± 7.2 | (17.53 ± 1.00)% |
> | Mistral-7B (Base) | 4 | 0.8 ± 0.8 | 319.8 ± 6.2 | (0.48 ± 0.46)% |
> |  |  |  |  |  |
> | Ministral-8B (SFT + RL) | 25 | 17.6 ±  1.7 | 183.2 ± 5.1 | (26.47 ± 1.01)% |
> | Ministral-8B (SFT-only) | 22 | 13.6 ± 3.6 | 201.4 ± 7.4 | (22.95 ± 2.39)% |
> | Ministral-8B (RL-only) | 3 | 2.6 ±  0.5 | 141.6 ± 3.8 | (27.05 ± 0.72)% |
> | Ministral-8B (Base) | 1 | 0.2 ± 0.4 | 293.4 ± 8.8 | (3.65 ± 0.86)% |
> |  |  |  |  |  |
> | Llama-3-8B-Instruct (SFT + RL) | 21 | 16.2 ± 1.3 | 121.2 ± 3.3 | (21.15 ± 0.75)%  |
> | Llama-3-8B-Instruct (SFT-only) | 20 | 15 ± 1.6 | 123.6 ± 3.2 | (21.27 ± 1.41)% |
> | Llama-3-8B-Instruct (RL-only) | 1 | 0.4 ± 0.5 | 164.2 ± 3 | (0.00 ± 0.00)% |
> | Llama-3-8B-Instruct (Base) | 0 | 0 | 217.6 ± 3.5 | (0.15 ± 0.41)% |
> |  |  |  |  |  |
> | Llama-3.1-8B-Instruct (SFT + RL) | 27 | 17.2 ± 2.3 | 183 ± 7.2 | (24.94 ± 1.74)% |
> | Llama-3.1-8B-Instruct (SFT-only) | 26 | 16.8 ± 1.1 | 190.2 ± 6.1 | (25.33 ± 1.05)% |
> | Llama-3.1-8B-Instruct (RL-only) | 20 | 13 ± 2.5 | 171.4 ± 3.6 | (31.61 ± 0.61)% |
> | Llama-3.1-8B-Instruct (Base) | 0 | 0 | 322 ± 2.6 | (0.52 ± 0.30)% |
> |  |  |  |  |  |
> | Qwen3-4B (SFT + RL) | 25 | 19.2 ± 1.3 | 183 ± 4.7 | (27.63 ± 0.99)% |
> | Qwen3-4B (SFT-only) | 25 | 16.4 ± 2.7 | 190 ± 4.1 | (24.16 ± 0.77)% |
> | Qwen3-4B (RL-only) | 1 | 0.4 ± 0.5 | 114.2 ± 5.6 | (0.00 ± 0.08)% |
> | Qwen3-4B (Base) | 10 | 2.2 ± 1.9 | 305.4 ± 8 | (2.74 ± 0.60)% |
> |  |  |  |  |  |
> | Qwen3-8B (SFT + RL) | 27 | 19 ± 3 | 183 ± 3 | (27.2 ± 1.2)% |
> | Qwen3-8B (SFT-only) | 25 | 13.0 ± 1.0 | 192 ± 7 | (23.6 ± 1.4)% |
> | Qwen3-8B (RL-only) | 19 | 11.8 ± 0.8 | 205 ± 5 | (25.2 ± 1.2)% |
> | Qwen3-8B (Base) | 8 | 2.6 ± 1.5 | 294.6 ± 2.3 | (4.43 ± 1.15)% |
>
> [1] https://huggingface.co/Qwen/Qwen3-4B
>
> [2] https://huggingface.co/mistralai/Mistral-7B-Instruct-v0.3
>
> [3] https://huggingface.co/mistralai/Ministral-8B-Instruct-2410
>
> [4] https://huggingface.co/meta-llama/Meta-Llama-3-8B-Instruct
>
> [5] https://huggingface.co/meta-llama/Llama-3.1-8B-Instruct
>
> [6] https://huggingface.co/PIPer-iclr
>
> [7] https://github.com/PIPer-iclr/PIPer
>
> [8] Wang, Zengzhi, et al. "Octothinker: Mid-training incentivizes reinforcement learning scaling." In the AI4Math Workshop, ICML25.
>
> [9] Gandhi, Kanishk, et al. "Cognitive behaviors that enable self-improving reasoners, or, four habits of highly effective stars." In COLM 2025.
>
> [10] Liu, Zichen, et al. "Understanding r1-zero-like training: A critical perspective." In COLM 2025.

---

### Official Review · Reviewer_RvYg · 2025-10-31

**Soundness:** 2
**Presentation:** 3
**Contribution:** 2
**Rating:** 2
**Confidence:** 5

**Summary:**

This paper proposes PIPER, a language model designed to configure terminal environments for successfully running projects and tasks. The authors fine-tune a smaller Qwen3 model using supervised fine-tuning (SFT) on training data distilled from a larger model of the same family. Given the high computational cost of executing generated bash scripts to verify correctness, the authors employ an LLM-as-a-Judge approach as a proxy for reward estimation during reinforcement learning (RL) training.

**Strengths:**

* The paper is clearly written and well-structured.
* The application of reinforcement learning to improve language models for environment configuration tasks is novel and practically relevant.
* **PIPER** achieves performance on par with larger models in benchmarks such as **Repo2Run** and **Terminal-Bench**, demonstrating promising efficiency gains.

**Weaknesses:**

### Overall

* There remains significant room for improvement: even the best-performing model (GPT-5) solves only **106/420** and **45/80** problems on **Repo2Run** and **Terminal-Bench**, respectively. Given that the larger **Qwen3-32B** model already performs poorly on these tasks (e.g., **29/329** on **EnvBench**), distilling from it is questionable. Typically, distillation is motivated by transferring the performance of a *highly capable* model to smaller ones, which does not seem justified in this setting.
* Using **LLM-as-a-Judge** as a proxy for script execution removes the verifiable reward component essential to reinforcement learning. It is unclear how accurate the judge model is at assessing correctness or identifying different error types. Although Figure 2 shows an increase in reward values during RL training, this does not translate to high absolute performance on **EnvBench**. This suggests that while Qwen3-8B is improving, its potential is constrained by the limited reliability of the feedback signal from the LLM-as-a-Judge. A systematic evaluation or profiling of the judge model’s accuracy is missing and should be included.
* Additionally, have the authors considered re-prompting the model with feedback from the environment when a setup attempt fails, and tasking it to iteratively refine the generated bash script? This strategy could be more effective than simply using pass@N, since both approaches involve multiple environment runs, but re-prompting may better guide the model toward the correct solution through targeted feedback rather than independent sampling.

---

### Data Collection

* It is unclear why the authors did not experiment with closed-source models such as **GPT-5** (or **Claude**) for collecting executable script data, even though these models are evaluated as baselines. Using more capable models for data generation could have produced higher-quality training data and potentially improved performance.

---

### Evaluation

* Given the high cost of executing model-generated shell scripts, **pass@1** would be a more suitable and interpretable metric for both in- and out-of-distribution benchmarks. What are the pass@1 numbers for all baselines and for PIPER on **Repo2Run** and **Terminal-Bench**? Such results would more clearly reveal each model’s ability to set up environments successfully on the first attempt.

---

### Generalization

* Although the authors defer exploration of other model families and scales to future work, an ablation or scaling analysis would be important to establish the general effectiveness of the proposed tuning method and to strengthen the paper’s contribution.
* The poor performance on **Terminal-Bench**, where PIPER underperforms even the base model, is insufficiently explained. One possible reason could be overfitting to the single-turn data distribution, resulting in the model only being effective on simpler, single-turn tasks from **Terminal-Bench**. The authors should discuss this and clarify the limitations of their approach.

**Questions:**

* See the Weaknesses Section.

---

> ### Author Response · Authors · 2025-11-20
> **Response to reviewer RvYg (part 1)**
>
> > Given that the larger **Qwen3-32B** model already performs poorly on these tasks (e.g., **29/329** on **EnvBench**), distilling from it is questionable. Typically, distillation is motivated by transferring the performance of a *highly capable* model to smaller ones, which does not seem justified in this setting.
> >
>
> **Response:**
> We respectfully disagree with the characterization that Qwen3-32B performs poorly on these tasks. As shown in Table 2, Qwen3-32B achieves 29 *pass@5* on EnvBench, which is competitive with strong closed-source models like GPT-4o (also 29 *pass@5*) and substantially better than GPT-4o-mini (15 *pass@5*), while the *avg@5* metric shows 16.2 successful setups compared to 19.0 for GPT-4o and 25.0 for GPT-5. While in absolute numbers this can be considered weak, it's still stronger than the student model and has a reasonably high score on the benchmark, making it a useful teacher.
>
> Furthermore, the relatively low absolute scores across all models, including frontier models like GPT-5, reflect the inherent difficulty of the EnvBench benchmark, which we discuss in Section 2. This complexity is precisely what makes the task interesting and relevant in our opinion, as it represents a genuine bottleneck in scaling execution-based software engineering benchmarks and training data collection.
>
> > Using **LLM-as-a-Judge** as a proxy for script execution removes the verifiable reward component essential to reinforcement learning. It is unclear how accurate the judge model is at assessing correctness or identifying different error types. […] A systematic evaluation or profiling of the judge model’s accuracy is missing and should be included.
> >
>
> **Response:**
> Firstly, we would like to thank the reviewer for raising this important question. To answer it, we conducted an additional study of the accuracy of our proposed LLM-as-a-Judge reward. For this, we ran the scripts from GPT-4o, Qwen3-32B, Qwen3-4B, Mistral-7B-Instruct, Qwen3-8B, Llama-3.1-8B-Instruct, and PIPer through both LLM-as-a-Judge and the evaluation pipeline, resulting in a total of 2229 scripts.
>
> To report the judge performance, we approach this as a classification task of predicting one of the three possible outcomes: (1) "failure": exit code != 0; (2) "has_issues": exit code = 0, >0 issues; (3) "success": exit code = 0, 0 issues. In such a case, the judge shows an accuracy of 0.59 and a macro F1 score of 0.42. We also studied the performance of the judge depending on the model and found that it's lower for the high-performing models (GPT-4o, Qwen3-32B, and PIPer), and higher for smaller base models (Qwen3-4B, Qwen3-8B, Mistral-7B-Instruct, Llama-3.1-8B-Instruct), which can explain why the judge was so useful in our training. We will include the full analysis in the camera-ready version.
>
> > Figure 2 shows an increase in reward values during RL training, this does not translate to high absolute performance on **EnvBench**. This suggests that while Qwen3-8B is improving, its potential is constrained by the limited reliability of the feedback signal from the LLM-as-a-Judge.
> >
>
> **Response:**
> We appreciate this observation and will clarify the relationship between reward improvements and task performance in the camera-ready version of the paper. The gains during training (shown in Figure 2) correlate with the increased performance on tests (shown in Table 2). We report the following gains over the base model: the combined approach (SFT+RL) improves pass@5 from 8 to 27 and avg@5 from 2.6 to 19; the RL-only approach improves pass@5 from 8 to 19 and avg@5 from 2.6 to 11.8. This shows that while the feedback can be of low fidelity, it effectively guides the model improvement.
>
> We acknowledge that the reward is approaching its maximum value (~0.75-0.9) while absolute performance remains below that of the frontier models, which could suggest either limitations in the reward signal's discrimination at higher performance levels or fundamental constraints of the single-turn scaffold we employ. We believe that to mitigate the latter, combining our approach with trajectory-level rewards for multi-turn scaffolds is a promising strategy, and outline this as a future research direction. And, to mitigate the former, we outline the future work of building stronger infrastructure for the high-fidelity rewards. However, we would like to stress that within the single-turn constraint, our method achieves performance competitive with much larger and more expensive models (matching GPT-4o and Qwen3-32B), suggesting that the LLM-as-a-Judge reward provides sufficiently reliable feedback for substantial capability improvements, even if further gains may require architectural changes beyond our current scope.

---

> ### Author Response · Authors · 2025-11-20
> **Response to reviewer RvYg (part 2)**
>
> > Additionally, have the authors considered re-prompting the model with feedback from the environment when a setup attempt fails, and tasking it to iteratively refine the generated bash script? This strategy could be more effective than simply using pass@N, since both approaches involve multiple environment runs, but re-prompting may better guide the model toward the correct solution through targeted feedback rather than independent sampling.
> >
>
> **Response:**
> We investigated the performance of several baselines in the Bash Agent scaffold [1] tailored for the environment setup task. It follows a widely used ReAct paradigm [2], providing the model with access to a Bash terminal with which it can interact iteratively. We use the original code and hyperparameters, allowing the models up to 30 iterations.
>
> We will include the full results in the camera-ready version, but we would like to note, that while this agentic approach improves the baseline model (Qwen3-8B) from 8 to 14 (pass@5) and from 2.6 to 4.8 (pass@1), PIPer still outperforms it on both metrics (27 pass@5 and 19 pass@1), while being times cheaper due to the iterative nature of the agent. The agentic setup notably boosts the GPT-4o model on the pass@5 metric (from 29 to 36), but lowers the pass@1 (from 19 to 17.4). As a result, while this work was focused on the single-step environment setup, we consider it is competitive with the agentic setup applied to the models of the same scale. We acknowledge the limitations of the current approach and the future line of work in Section 7.
>
> > It is unclear why the authors did not experiment with closed-source models such as **GPT-5** (or **Claude**) for collecting executable script data, even though these models are evaluated as baselines. Using more capable models for data generation could have produced higher-quality training data and potentially improved performance.
> >
>
> **Response:**
> We appreciate this question regarding our choice of teacher model for SFT data collection. During the preparation of this study, we experimented with various sources of SFT data, including different filtering policies. We observed similar final performance after completing the SFT training pipeline, suggesting that the choice of teacher model has a limited impact on the capabilities of the final trained model, primarily affecting the format of the model output.
>
> We chose to report results using only open-source models (specifically Qwen3-32B) due to potential legal concerns: training models on trajectories generated by OpenAI or Anthropic models is restricted by their terms of service [3, 4], which limits the reproducibility and broader applicability of our work. Moreover, distillation from models not controlled by the community can further restrict reproducibility since such a model can be deprecated by its vendor. By using open-source models exclusively, we ensure that our approach can be freely adopted, reproduced, and extended by other researchers without legal constraints, and all our training data, code, and model checkpoints can be publicly released.
>
> [1] Eliseeva, A., Kovrigin, A., Kholkin, I., Bogomolov, E., & Zharov, Y. EnvBench: A Benchmark for Automated Environment Setup. In ICLR 2025 Third Workshop on Deep Learning for Code.
>
> [2] Yao, S., Zhao, J., Yu, D., Du, N., Shafran, I., Narasimhan, K. R., & Cao, Y. (2022, October). ReAct: Synergizing Reasoning and Acting in Language Models. In ICLR 2022.
>
> [3] https://openai.com/policies/services-agreement/
>
> [4] https://support.claude.com/en/articles/12326764-can-i-use-my-outputs-to-train-an-ai-model

---

> ### Author Response · Authors · 2025-11-20
> **Response to reviewer RvYg (part 3)**
>
> > Given the high cost of executing model-generated shell scripts, **pass@1** would be a more suitable and interpretable metric for both in- and out-of-distribution benchmarks. What are the pass@1 numbers for all baselines and for PIPER on **Repo2Run** and **Terminal-Bench**? Such results would more clearly reveal each model’s ability to set up environments successfully on the first attempt.
> >
>
> **Response:**
> We appreciate this suggestion regarding pass@1 metrics. In our study, we chose to report pass@5 for Repo2Run and pass@10 for Terminal-Bench, which provides a fairer and stable comparison across evaluated models[5]. This happens due to the high variance in model outputs and success rates, especially on out-of-distribution benchmarks. However, we report pass@1 results here and will add them to the paper:
>
> | Model | Repo2Run | Terminal-Bench |
> |-------|----------|----------------|
> |  | avg pass@1| pass@1 |
> | GPT-5 | 41.2 ± 22.4 | 20 |
> | GPT-4o | 32.0 ± 10.7 | 9 |
> | GPT-4o-mini | 54.6 ± 2.9 | 9 |
> | Qwen3-32B | 41.6 ± 4.3 | 9 |
> | Qwen3-14B | 31.2 ± 4.7 | 5 |
> | Qwen3-8B | 9.6 ± 2.5| 3 |
> | PIPer | 79.2 ± 4.4| 4 |
> | PIPer-RL-only | 52.0 ± 4.6 | 4 |
> | PIPer-SFT-only | 62.0 ± 2.3 | 1 |
>
> *NB! We report Terminal-Bench pass@1 results from the original evaluation suite. In the camera-ready version of the paper, we will include average values.*
>
> However, for smaller on-device-sized models employed in automated pipelines (similarly to how it is used in [6,7,8] and many more), using pass@N becomes a practically viable and cost-effective approach since users can run multiple attempts at essentially no cost on their local hardware.
>
> > Although the authors defer exploration of other model families and scales to future work, an ablation or scaling analysis would be important to establish the general effectiveness of the proposed tuning method and to strengthen the paper’s contribution.
> >
>
> **Response:**
> We concur with the reviewer, and are performing the additional ablation studies. We expect to report on the results within a couple of days. However, based on the preliminary results we have, we do not see any change in the model behavior.
>
> > The poor performance on **Terminal-Bench**, where PIPER underperforms even the base model, is insufficiently explained. One possible reason could be overfitting to the single-turn data distribution, resulting in the model only being effective on simpler, single-turn tasks from **Terminal-Bench**. The authors should discuss this and clarify the limitations of their approach.
> >
>
> **Response:**
> We acknowledge that the poor performance on Terminal-Bench, particularly for SFT-only, deserves an expanded discussion. We concur with the proposed explanation, and will expand the discussion in the camera-ready version of the paper. We also note that the trained model still demonstrates better capability to address terminal-related tasks outside the environment setup domain, indicating that the learned scripting knowledge partially transfers even when the interaction domain differs.
>
> [5] Paul, Debalina Ghosh, Hong Zhu, and Ian Bayley. "Benchmarks and metrics for evaluations of code generation: A critical review." 2024 IEEE International Conference on Artificial Intelligence Testing (AITest). IEEE, 2024.
>
> [6] Jimenez, Carlos E., et al. "Swe-bench: Can language models resolve real-world github issues?." In ICLR 2024.
>
> [7] Badertdinov, Ibragim, et al. "SWE-rebench: An Automated Pipeline for Task Collection and Decontaminated Evaluation of Software Engineering Agents." In NeurIPS 2025.
>
> [8] Vergopoulos, Konstantinos, Mark Niklas Müller, and Martin Vechev. "Automated benchmark generation for repository-level coding tasks." In DL4C@ICLR'25 and FMWild@ICLR'25.

---

> > ### Comment · Reviewer_RvYg · 2025-11-26
> >
> > Thank you for the detailed rebuttal. I appreciate the authors’ effort in addressing several of my concerns, particularly regarding pass@1 and comparisons with multi-turn agentic baselines. The proposed single-turn model for on-device environment setup is a strong result relative to larger off-the-shelf models. However, a few key concerns remain:
> >
> > - Trajectory distribution and data collection. Although PIPer’s parity with GPT-4o follows from using Qwen3-32B—which performs similarly to GPT-4o—as a teacher model, the absolute performance remains low. In its current form, the trajectory dataset reflects the limitations of the teacher itself. A broader contribution to the ICLR community would explore trajectory collection strategies that do not inherit performance ceilings from the teacher model used to generate them.
> > - LLM-as-a-Judge reliability and verifiable feedback. The LLM-based reward model achieves a Macro-F1 of only 0.42, indicating that it cannot reliably distinguish among the three execution outcomes. Because its reward signal is therefore a non-verifiable proxy (rather than grounded in actual script execution), the student model is susceptible to reward hacking. This is consistent with Figure 2, where the model receives high rewards during training/validation, yet Table 2 shows poor execution-grounded results on the test split. While I understand the actually evaluating the bash scripts would be expensive slowing down the RL training, a useful contribution would be to come up with reward functions that are not bounded by proxy unreliable feedback but are more towards representing true task success.
> >
> >
> > Based on this, I have updated my score accordingly.

---

> > > ### Author Response · Authors · 2025-11-26
> > > **Response to reviewer RvYg (follow-up 1)**
> > >
> > > **We thank Reviewer RvYg for the thoughtful feedback and the score update.** We agree with the reviewer that these contributions constitute a strong follow-up to the work and will expand Section 7 accordingly. Overall, we maintain that the current contribution of the paper lies in democratizing the environment setup task by tuning small models to perform on par with the closed-source ones. We release the dataset, evaluation results, trained models, and full training suite (including all prompts and configs), which can serve as a foundation for the broader research directions.
> > >
> > > We appreciate the reviewer's patience while we conducted the additional experiments on model families and scales that were requested. We now return to the question we promised to address in our earlier response.
> > >
> > > > Although the authors defer exploration of other model families and scales to future work, an ablation or scaling analysis would be important to establish the general effectiveness of the proposed tuning method and to strengthen the paper’s contribution.
> > > >
> > >
> > > **Response:** To thoroughly address this question and demonstrate the robustness of the proposed method, we have deployed it across multiple model families and sizes, focusing on the small, on-device-sized models. Specifically, we experimented with Qwen3-4B [1], Mistral-7B [2], Ministral-8B [3], Llama-3-8B-Instruct [4], and Llama-3.1-8B-Instruct [5]. We applied the same hyperparameters reported in the paper across all models.
> > >
> > > We attach the resulting table in the next comment and will use the 1-page extension of the paper to report and discuss this ablation study. All new trained model checkpoints, along with their associated training dynamics plots, have been published on our HuggingFace [6] and GitHub repository [7] for full transparency and reproducibility.
> > >
> > > Due to the limited resources, we haven’t tuned the hyperparameters of our method for each of the models under investigation, and we acknowledge that the results of each individual model can be further improved. However, across all model families and sizes, we observe behaviour similar to that of the Qwen3-8B presented in the paper. The two models that stand out are Qwen3-4B and Llama-3-8B-Instruct. Neither of them improved during the RL-only phase, but both show improvement during the SFT+RL phase. For the Qwen3-4B, we hypothesize that this may be due to the base model's inability to generate well-formatted scripts, thereby hindering RL training. And for Llama-3-8B-Instruct, it is a known limitation [8, 9, 10]. Driven by the results of various model sizes, we hypothesize that one important hyperparameter of the proposed method is the balance between SFT and RL. While the smaller models may require more SFT examples, the larger ones may be impeded by this phase. We leave a thorough research of this dependency for further work.
> > >
> > > [1] https://huggingface.co/Qwen/Qwen3-4B
> > >
> > > [2] https://huggingface.co/mistralai/Mistral-7B-Instruct-v0.3
> > >
> > > [3] https://huggingface.co/mistralai/Ministral-8B-Instruct-2410
> > >
> > > [4] https://huggingface.co/meta-llama/Meta-Llama-3-8B-Instruct
> > >
> > > [5] [https://huggingface.co/meta-llama/Llama-3.1-8B-Instruct](https://huggingface.co/meta-llama/Meta-Llama-3-8B-Instruct)
> > >
> > > [6] https://huggingface.co/PIPer-iclr
> > >
> > > [7] https://github.com/PIPer-iclr/PIPer
> > >
> > > [8] Wang, Zengzhi, et al. "Octothinker: Mid-training incentivizes reinforcement learning scaling." In the AI4Math Workshop, ICML25.
> > >
> > > [9] Gandhi, Kanishk, et al. "Cognitive behaviors that enable self-improving reasoners, or, four habits of highly effective stars." In COLM 2025.
> > >
> > > [10] Liu, Zichen, et al. "Understanding r1-zero-like training: A critical perspective." In COLM 2025.

---

> > > ### Author Response · Authors · 2025-11-26
> > > **Response to reviewer RvYg (follow-up 2)**
> > >
> > > | **Model** | **pass@5 #Success ↑** | **avg@5 #Success ↑** | **avg@5 #Failed ↓** | **avg@5 avgFixRate ↑** |
> > > | --- | --- | --- | --- | --- |
> > > | Mistral-7B (SFT + RL) | 22 | 16.6 ± 1.5 | 169.8 ± 4.2 | (29.42 ± 0.71)% |
> > > | Mistral-7B (SFT-only) | 18 | 12.4 ± 1.7 | 211.8 ± 5.8 | (21.56 ± 0.83)% |
> > > | Mistral-7B (RL-only) | 8 | 3.4 ± 0.5 | 181.6 ± 7.2 | (17.53 ± 1.00)% |
> > > | Mistral-7B (Base) | 4 | 0.8 ± 0.8 | 319.8 ± 6.2 | (0.48 ± 0.46)% |
> > > |  |  |  |  |  |
> > > | Ministral-8B (SFT + RL) | 25 | 17.6 ±  1.7 | 183.2 ± 5.1 | (26.47 ± 1.01)% |
> > > | Ministral-8B (SFT-only) | 22 | 13.6 ± 3.6 | 201.4 ± 7.4 | (22.95 ± 2.39)% |
> > > | Ministral-8B (RL-only) | 3 | 2.6 ±  0.5 | 141.6 ± 3.8 | (27.05 ± 0.72)% |
> > > | Ministral-8B (Base) | 1 | 0.2 ± 0.4 | 293.4 ± 8.8 | (3.65 ± 0.86)% |
> > > |  |  |  |  |  |
> > > | Llama-3-8B-Instruct (SFT + RL) | 21 | 16.2 ± 1.3 | 121.2 ± 3.3 | (21.15 ± 0.75)%  |
> > > | Llama-3-8B-Instruct (SFT-only) | 20 | 15 ± 1.6 | 123.6 ± 3.2 | (21.27 ± 1.41)% |
> > > | Llama-3-8B-Instruct (RL-only) | 1 | 0.4 ± 0.5 | 164.2 ± 3 | (0.00 ± 0.00)% |
> > > | Llama-3-8B-Instruct (Base) | 0 | 0 | 217.6 ± 3.5 | (0.15 ± 0.41)% |
> > > |  |  |  |  |  |
> > > | Llama-3.1-8B-Instruct (SFT + RL) | 27 | 17.2 ± 2.3 | 183 ± 7.2 | (24.94 ± 1.74)% |
> > > | Llama-3.1-8B-Instruct (SFT-only) | 26 | 16.8 ± 1.1 | 190.2 ± 6.1 | (25.33 ± 1.05)% |
> > > | Llama-3.1-8B-Instruct (RL-only) | 20 | 13 ± 2.5 | 171.4 ± 3.6 | (31.61 ± 0.61)% |
> > > | Llama-3.1-8B-Instruct (Base) | 0 | 0 | 322 ± 2.6 | (0.52 ± 0.30)% |
> > > |  |  |  |  |  |
> > > | Qwen3-4B (SFT + RL) | 25 | 19.2 ± 1.3 | 183 ± 4.7 | (27.63 ± 0.99)% |
> > > | Qwen3-4B (SFT-only) | 25 | 16.4 ± 2.7 | 190 ± 4.1 | (24.16 ± 0.77)% |
> > > | Qwen3-4B (RL-only) | 1 | 0.4 ± 0.5 | 114.2 ± 5.6 | (0.00 ± 0.08)% |
> > > | Qwen3-4B (Base) | 10 | 2.2 ± 1.9 | 305.4 ± 8 | (2.74 ± 0.60)% |
> > > |  |  |  |  |  |
> > > | Qwen3-8B (SFT + RL) | 27 | 19 ± 3 | 183 ± 3 | (27.2 ± 1.2)% |
> > > | Qwen3-8B (SFT-only) | 25 | 13.0 ± 1.0 | 192 ± 7 | (23.6 ± 1.4)% |
> > > | Qwen3-8B (RL-only) | 19 | 11.8 ± 0.8 | 205 ± 5 | (25.2 ± 1.2)% |
> > > | Qwen3-8B (Base) | 8 | 2.6 ± 1.5 | 294.6 ± 2.3 | (4.43 ± 1.15)% |

---

### Official Review · Reviewer_sq94 · 2025-11-01

**Soundness:** 2
**Presentation:** 2
**Contribution:** 2
**Rating:** 2
**Confidence:** 4

**Summary:**

This paper addresses the task of automated software environment setup. The authors propose a two-stage training pipeline to fine-tune a small language model (Qwen3-8B) for this task. The method first uses supervised fine-tuning (SFT) on a dataset of executable scripts distilled from a larger "teacher" model (Qwen3-32B). This is followed by a reinforcement learning (RL) stage using a verifiable reward (RLVR), where the reward signal is provided by an "LLM-as-a-Judge" as a proxy for true script execution. The authors evaluate their final model on EnvBench-Python, Repo2Run, and Terminal-Bench, showing that their 8B model can achieve performance comparable to much larger models like GPT-4o on the primary benchmark.

**Strengths:**

1. The paper tackles a practical problem in software engineering. Automating environment setup is a well-known bottleneck, and a robust solution would have a significant real-world impact.
2. The core result—that a specialized 8B parameter model can match the performance of SOTA models like GPT-4o and a 32B model—is good.
3. The authors state their intention to release code, models, and generated scripts, which is commendable and valuable for the community.

**Weaknesses:**

1. The primary weakness of this paper is its limited contribution to methodology. The proposed "SFT + RLVR" pipeline is a well-established recipe. The SFT stage is standard knowledge distillation, and the RL stage applies known algorithms (REINFORCE++) with an LLM-as-a-Judge reward. This technique has been widely used. The paper presents a successful application of this pipeline to a new domain (environment setup) but does not introduce new learning principles or techniques that would be of broad interest to the ICLR community.
2. The paper's writing is clear but largely descriptive. It reads as a straightforward "we applied method X to problem Y and got good results". The narrative fails to build a strong case for why this specific application (environment setup) poses unique or interesting challenges that necessitate new ML insights, or what the broader ML community can learn from this specific application.
3. The paper's central claim of being "on par" with much larger models (GPT-4o, Qwen3-32B) is based on a flawed "apples-to-oranges" comparison. The authors compare their highly specialized, tuned 8B model against general-purpose, untuned (zero-shot) baselines. This comparison only proves the known fact that task-specific tuning is effective. A fair and more insightful comparison would require comparing against other tuned models. For instance, what is the performance of the Qwen3-32B "teacher" model after it also undergoes the SFT+RLVR training? The current setup compares a tuned small model to an untuned large model, which inflates the perceived relative performance of the 8B model.

**Questions:**

1. Given that the SFT+RLVR pipeline is standard, what do the authors believe is the core contribution? Did the environment setup task require specific, novel modifications to the RL algorithm or reward formulation that would be generalizable to other structured generation tasks?
2. The SFT data was filtered to include only successful executable scripts. Do the authors worry this introduces a bias? The model may not learn how to recover from or debug failed scripts, as it may not have seen such "failure-recovery" patterns in the SFT data. Did you experiment with including any "near-miss" or "productively-failed" scripts in the SFT dataset?

---

> ### Author Response · Authors · 2025-11-20
> **Response to reviewer sq94**
>
> > Given that the SFT+RLVR pipeline is standard, what do the authors believe is the core contribution? Did the environment setup task require specific, novel modifications to the RL algorithm or reward formulation that would be generalizable to other structured generation tasks?
> >
>
> **Response:**
> Thank you for this important question about our contributions. We want to emphasize that ICLR explicitly encourages work of this nature. The call for papers states:
>
> > We consider a broad range of subject areas including [...] **applications** in vision, audio, speech, language, music, robotics, games, healthcare, biology, sustainability, economics, ethical considerations in ML, and others.
> >
>
> Our work falls within this scope: we apply existing techniques to a novel, practically important domain where they have not been systematically explored. As noted by the reviewers, the problem is both practical and actual for the community. The complexity of the task is highlighted by the results on the EnvBench: GPT-5 achieves only 43/329 success (13%), and prior work reports that automated approaches struggle to scale beyond small curated datasets [1,2].
>
> Our core contributions, therefore, are: (1) a practical stepping stone for scaling the training of agents on live environments, (2) a demonstration that lightweight LLM-as-a-Judge rewards can substitute expensive containerized execution during RLVR training, and (3) a rigorous study of the results and problems that arise before the models on the task of environment setup.
>
> > The SFT data was filtered to include only successful executable scripts. Do the authors worry this introduces a bias? The model may not learn how to recover from or debug failed scripts, as it may not have seen such "failure-recovery" patterns in the SFT data. Did you experiment with including any "near-miss" or "productively-failed" scripts in the SFT dataset?
> >
>
> **Response:**
> We would like to stress that our approach is specifically designed for a single-turn, non-interactive scaffold. We acknowledge this limitation in Section 7, and the rest of the paper is set within this framework. In our setup, including failed scripts during SFT would not provide an actionable learning signal, as the model has no mechanism to observe or respond to failures. We acknowledge the high relevance of the question to the broader topic of distillation for the task of environment setup, but consider it outside the scope of this paper.
>
> > The paper's central claim of being "on par" with much larger models (GPT-4o, Qwen3-32B) is based on a flawed "apples-to-oranges" comparison. The authors compare their highly specialized, tuned 8B model against general-purpose, untuned (zero-shot) baselines. This comparison only proves the known fact that task-specific tuning is effective. A fair and more insightful comparison would require comparing against other tuned models. For instance, what is the performance of the Qwen3-32B "teacher" model after it also undergoes the SFT+RLVR training?
> >
>
> **Response:**
> While we agree that the results of stronger models under the same training conditions are interesting, we disagree with the judgment that the presented comparison is incorrect. The practice of comparing a small tuned model to the performance of off-the-shelf generalist models is widely adopted in the community. This comparison highlights the practical implications of the method—saving money while maintaining performance. Since the aim of the work is to expand the abilities of small models, training large models was considered out of scope for this work. Finally, the performance of a small model is compared to a large generalist model, and the result of the comparison wasn’t as assured, given the fast pace of improvements, especially concerning the software engineering area.
>
> [1] Badertdinov, Ibragim, et al. "SWE-rebench: An Automated Pipeline for Task Collection and Decontaminated Evaluation of Software Engineering Agents." *NeurIPS 2025*.
>
> [2] Kuang, Jiayi, et al. "Process-Level Trajectory Evaluation for Environment Configuration in Software Engineering Agents." *arXiv preprint arXiv:2510.25694* (2025).

---

### Official Review · Reviewer_fgA7 · 2025-11-01

**Soundness:** 3
**Presentation:** 3
**Contribution:** 4
**Rating:** 8
**Confidence:** 4

**Summary:**

This paper presents PIPER, a specialized model for automating environment setup in software engineering projects. The authors address a persistent challenge in software development where configuring systems to work with specific repositories requires significant manual effort, which also limits the scalability of execution-based benchmarks for SE research. They develop their approach by combining supervised fine-tuning on scripts generated by larger models with RLVR, using a lightweight LLM-as-a-Judge reward function to avoid the computational overhead of executing scripts in isolated containers. The key contribution is demonstrating that their fine-tuned Qwen3-8B model—small enough to run on consumer hardware—achieves performance comparable to much larger models like Qwen3-32B and GPT-4o on the EnvBench-Python benchmark, showing more than 9× improvement over the base model. Through careful analysis of failure patterns in model-generated scripts and evaluation across multiple benchmarks including Repo2Run and Terminal-Bench, they show that their training approach develops genuine scripting capabilities that generalize beyond the training distribution, offering researchers and developers a more accessible and cost-effective solution for automated environment setup.

**Strengths:**

1. This paper tries to solve a very important problem, environment setup, which is useful for building coding environments for coding agents. This is a pain point in many questions. The work has strong practical significance for the SE research community.
2. The key insight is to validate environment setup through LLM-as-a-Judge in reinforcement learning. In this way, it avoid execution overhead in containers. This can significanly reducing the training time and infrastructure requirements, while still preseve the debugging interpretability.
3. While the training pipeline itself is conventional, the analysis in this paper is very detailed. The paper evaluates across three diverse benchmarks (EnvBench, Repo2Run, Terminal-Bench) with different success criteria (static analysis, test collection, task completion), demonstrating the robustness of findings beyond a single evaluation protocol. The ablation studies clearly show both SFT and RL stages contribute meaningfully.

**Weaknesses:**

1. No validation of LLM-as-a-Judge reward wuality. The LLM-as-a-Judge is the core innovation, yet there's no empirical evidence it produces accurate predictions.
2. The claim of 'on-device' is confusing. This paper is not about on-device training or on-device inference.

**Questions:**

1. Could you please provide correlation analysis between judge predictions and actual execution outcomes on a validation set? It would be interesting to see the false positive and false negatives. A detailed error analysis will be appriciated.
2. Why did GPT-4o and GPT-4o-mini fail as judges? Is there any detailed explanation?
3. How sensitive is performance to the judge prompt and criteria?

---

> ### Author Response · Authors · 2025-11-20
> **Response to reviewer fgA7**
>
> > Could you please provide correlation analysis between judge predictions and actual execution outcomes on a validation set? It would be interesting to see the false positive and false negatives. A detailed error analysis will be appriciated.
> >
>
> **Response:**
> Thank you for bringing this important question to our attention. To answer it, we conducted an additional study of the accuracy of our proposed LLM-as-a-Judge reward. For this, we ran the scripts from GPT-4o, Qwen3-32B, Qwen3-4B, Mistral-7B-Instruct, Qwen3-8B, Llama-3.1-8B-Instruct, and PIPer through both LLM-as-a-Judge and the evaluation pipeline, resulting in a total of 2229 scripts.
>
> To report the judge performance, we approach this as a classification task of predicting one of the three possible outcomes: (1) "failure": exit code != 0; (2) "has_issues": exit code = 0, >0 issues; (3) "success": exit code = 0, 0 issues. In such a case, the judge shows an accuracy of 0.59 and a macro F1 score of 0.42. We also studied the performance of the judge depending on the models and found that it's lower for the high-performing models (GPT-4o, Qwen3-32B, and PIPer), and higher for smaller base models (Qwen3-4B, Qwen3-8B, Mistral-7B-Instruct, Llama-3.1-8B-Instruct), which can explain why the judge was so useful in our training. We will include the full analysis in the camera-ready version.
>
> > Why did GPT-4o and GPT-4o-mini fail as judges? Is there any detailed explanation?
> >
>
> **Response:**
> To answer this question, we ran LLM-as-a-Judge with GPT-4.1, GPT-4o, and GPT-4o-mini as backbones for GPT-4o, Qwen3-32B, Qwen3-8B, and PIPer models as generators, resulting in a total of 1279 scripts. To investigate the judge's performance, we use the same classification task as in the previous answer.
>
> | LLM-as-a-Judge Backbone  | Failure F1 |   Has Issues F1 |   Success F1 |
> |:-----------|------------:|---------:|----------:|
> | GPT-4.1    |   0.53  | 0.49 |  0.09 |
> | GPT-4o    |   0.53  | 0.30 | 0.08  |
> | GPT-4o-mini |   0.34  | 0.15 | 0.09  |
>
>
> We observe that GPT-4.1 dominates other judge models in terms of quality. While GPT-4o performs similarly in identifying Failure scripts, it is substantially worse for Has Issues, which is a relatively frequent outcome in our evaluation runs (the considered set of scripts contains 878 Failure, 351 Has Issues, and 50 Success samples). We hypothesize that this explains the difference in the final performance of the trained models.
>
> > How sensitive is performance to the judge prompt and criteria?
> >
>
> **Response:**
> While we fully agree that this is an interesting question worthy of an investigation, we haven’t conducted it. We have explored different configurations and prompts, but only to the extent necessary to make our method work, and haven’t investigated the full breadth of this question.
>
> > The claim of 'on-device' is confusing. This paper is not about on-device training or on-device inference.
> >
>
> **Response:**
> Thank you for this question. We will further clarify this in the paper. By "on-device" we mean "on-device-sized" models that can run on consumer hardware (e.g., single consumer GPU, or no GPU at all) and be incorporated into local workflows (as defined in line 055 of the manuscript). This definition emphasizes the usage accessibility rather than the cheapness of training. Our goal is to democratize the environment setup by enabling local deployment without expensive API calls, while keeping competitive performance.

---

### Author Response · Authors · 2025-11-20
**Official Comment by the Authors**

We sincerely thank all reviewers for their thorough evaluation and constructive feedback. We are encouraged that Reviewer fgA7 recognized our work as addressing a very important problem with strong practical significance for the SE research community, and that all reviewers acknowledge the practical relevance and clear presentation of our work.

We have carefully considered concerns raised across the reviews. We provide detailed responses to each reviewer's specific questions in corresponding threads. To address some of the requests, we are currently employing our method to train additional models, which requires a noticeable amount of time. We will share the results within the next few days.

---

### Author Response · Authors · 2025-12-02
**Final Authors' Remark**

We sincerely thank all reviewers for their thorough evaluation and constructive feedback on our submission. Your insights have been invaluable in helping us strengthen our work.

In response to your feedback, we have conducted the following additional work:

- **Model Sensitivity Study** (detailed in responses to Reviewers RvYg and F4KY): We trained and evaluated our method across five additional model families and sizes (Qwen3-4B, Mistral-7B, Ministral-8B, Llama-3-8B-Instruct, and Llama-3.1-8B-Instruct), demonstrating the robustness of our approach. All checkpoints and plots with training dynamics are publicly available.
- **LLM-as-a-Judge Analysis** (detailed in responses to Reviewers fgA7 and RvYg): We performed a comprehensive evaluation of our LLM-as-a-Judge reward across 2229 scripts from 7 different models, analyzing its accuracy in classifying the expected evaluation outcome and investigating why different judge backbones perform differently and how our main judge’s performance differs across models.
- **Agentic Baseline Comparison** (detailed in responses to Reviewers RvYg and F4KY): We implemented and evaluated multi-turn agentic baselines using the Bash Agent scaffold with ReAct paradigm, showing that our single-turn approach remains competitive while being more cost-efficient.
- **Improving Results Clarity** (detailed in responses to Reviewers RvYg and F4KY): We provided comprehensive pass@1 results and per-repository success matrices for enhanced transparency, and extended our replication package with runtime/cost breakdowns. All aforementioned data is publicly available.

We have addressed each reviewer's specific concerns in the individual response threads. We will utilize the one-page extension to incorporate the improvements into the camera-ready version.

We would like to note that these additions do not alter the storyline or contributions of our paper, but rather further strengthen the results and their presentation.

Thank you again for your constructive engagement with our work!

---

### Meta-Review · Area_Chair_nShR · 2026-01-07

**Summary:**

The paper studies an interesting problem of automated environment setup with LLM agents and motivates its importance for automated and scalable data curation for training general purpose SWE agents. Scalable data curation of executable environments for training SWE agents currently requires significant manual effort which is a bottleneck. The paper received mixed reviews of (8,2,2,4) before the discussion period. While the reviewers appreciated the rich analysis, most reviewers noted that the limited novelty of proposed training (standard SFT + RL). Also while the paper is motivated with need for scalable data curation (L034-L049) there are no results to show that the proposed method impacts downstream SWE data curation and training. Furthermore, more recent automated data curation methods e.g., SWE-smith, R2EGym are not discussed yet. The paper discusses an interesting problem but in its current manuscript, it may benefit from another round of peer-review.

**Reviewer Concerns:**

See above.

**Reviewer Scores:**

The initial ratings were 8,2,2,4. One reviewer (who gave 2) methioned they would increase the score after seeing the rebuttal. Both unfortunately didn't happen. But overall, AC thinks that the tune of this paper is negative as discussed in the meta-review.

---

### Decision · Program_Chairs · 2026-01-26

Reject